# Imaging Cataract-Specific Peptides in Human Lenses

**DOI:** 10.3390/cells11244042

**Published:** 2022-12-14

**Authors:** Kevin L. Schey, Zhen Wang, Kristie L. Rose, David M. G. Anderson

**Affiliations:** Department of Biochemistry and Mass Spectrometry Research Center, Vanderbilt University School of Medicine, Nashville, TN 37232, USA

**Keywords:** imaging mass spectrometry, ocular lens, protein degradation, cataract

## Abstract

Age-related protein truncation is a common process in long-lived proteins such as proteins found in the ocular lens. Major truncation products have been reported for soluble and membrane proteins of the lens, including small peptides that can accelerate protein aggregation. However, the spatial localization of age-related protein fragments in the lens has received only limited study. Imaging mass spectrometry (IMS) is an ideal tool for examining the spatial localization of protein products in tissues. In this study we used IMS to determine the spatial localization of small crystallin fragments in aged and cataractous lenses. Consistent with previous reports, the pro-aggregatory αA-crystallin 66–80 peptide as well as αA-crystallin 67–80 and γS-crystallin 167–178 were detected in normal lenses, but found to be increased in nuclear cataract regions. In addition, a series of γS-crystallin C-terminal peptides were observed to be mainly localized to cataractous regions and barely detected in transparent lenses. Other peptides, including abundant αA3-crystallin peptides were present in both normal and cataract lenses. The functional properties of these crystallin peptides remain unstudied; however, their cataract-specific localization suggests further studies are warranted.

## 1. Introduction

Age-related nuclear cataract (ARNC) is the most common form of age-related lens opacification and is a leading cause of visual impairment [1]. Lens protein aggregation and the resulting scattering of light causes the opacification and multiple mechanisms have been proposed to lead to protein modification. Importantly, lens proteins are long-lived proteins (LLPs) and they can undergo many post-translational modifications over a lifetime that spans many decades [2,3,4,5]. Thus, the ocular lens is an ideal tissue to examine such changes given that the proteins localized in the central lens nucleus are as old as the donor. Modifications that have been characterized in aged human lenses include: deamidation, truncation, isomerization, racemization, and crosslinking [4,5,6,7,8]. An important aspect of lens protein modification is the specific location where such modifications occur as this information helps to decipher if modifications are age-related. Furthermore, identification of cataract-specific modifications may suggest potential cataractogenic mechanisms. Thus, spatially resolved studies are necessary to link lens age-related protein modifications to the lens nucleus and nuclear cataract.

Imaging mass spectrometry (IMS) is an imaging technology that can provide spatial localization for proteins, peptides, lipids, and metabolites with high spatial resolution [9,10]. IMS has been applied to lens tissue to define the location of lens proteins and their modified forms in human lenses as a function of age [4,11,12]. Based on these studies, lens protein truncation begins early in life and leads to highly truncated protein forms in older nuclear fiber cells. Indeed, a large number of crystallin peptides have been discovered in aged human lenses [13,14]. One peptide, αA-crystallin 66–80, has been shown to induce protein aggregation [15], suggesting that age-related truncation products can have deleterious effects in the lens. Furthermore, IMS analysis of a guinea pig model of ARNC showed that protein truncation in the lens nucleus was accelerated with hyperbaric oxygen (HBO) treatment [16]. Thus, knowledge of cataract-specific protein truncation products would inform on potentially cataractogenic properties of such peptides.

In this study, we employed IMS to image crystallin peptides in aged and cataractous human lenses. Specific peptides were reproducibly observed in nuclear cataract regions while not observed in age-matched transparent lenses. Other peptides were excluded from cataractous regions.

## 2. Materials and Methods

### 2.1. Tissue Preparation for Imaging Mass Spectrometry (IMS)

Human lenses were acquired from National Disease Research Interchange (NDRI) (Philadelphia, PA, USA), the South Carolina Lions Eye Bank (Charleston, SC, USA) or as a gift from Dr. Donita Garland (Ocular Genomics Institute at Massachusetts Eye and Ear, Harvard University, Cambridge, MA, USA). All lenses were stored at −80 °C until use. Information related to lenses used in this study can be found in the Appendix A. Each lens was sectioned equatorially to 10 μm thickness using a CM 3050 cryostat (Leica CM 3050S, Leica Microsystems Inc., Bannockburn, IL, USA). Tissue sections were thaw-mounted on conductive Indium-Tin-Oxide (ITO) coated slides (Delta Technologies, Loveland, CO, USA) and vacuum desiccated for 30 min. To remove lipids, the sections were washed twice with 70% ethanol for 1 min followed by washing with 100% ethanol for 1 min. The sections were then dried in a vacuum desiccator. To enable matrix-assisted laser desorption ionization (MALDI), tissue sections were spray-coated with 2,5-dihydroxybenzoic acid (DHB) matrix (20 mg/mL in 70% ACN containing 0.1% TFA) using a TM sprayer (HTX Technologies, LLC, Carrboro, NC, USA). The TM Sprayer was operated using the following settings: flow rate 0.1 mL/min, nozzle velocity of 700 mm/min, nozzle temperature at 85 °C, track spacing at 3 mm, number of passes of 8 and nitrogen pressure 9.5 psi. After matrix application, tissue sections were rehydrated at 85 °C for 3 min in a sealed chamber containing 50 µL of 50 mM acetic acid.

### 2.2. MALDI Data Acquisition

MALDI imaging mass spectrometry experiments were performed using a Bruker Rapiflex Tissuetyper (Bruker Daltonics, Billerica, MA, USA) equipped with a Smartbeam 3D 10 kHz 355 nm Nd:YAG laser™ 3D laser. *m*/*z* calibration was accomplished via a linear external calibration using a mix of bovine insulin, equine cytochrome c, bovine ubiquitin I, and equine myoglobin (Bruker Daltonik, Billerica, MA, USA). Mass spectrometric analyses were performed in the linear, positive ionization mode over a mass-to-charge (*m*/*z*) range of *m*/*z* 1200 to *m*/*z* 20,000. The ionization laser was scanned across the tissue surface with a raster step size of 60 μm and spectra from 2000 laser shots per pixel (tissue position) were averaged to produce a spectrum. Low molecular weight crystallin fragments were also analyzed in a positive ion reflector MS mode, to improve mass resolution, over an *m*/*z* range of *m*/*z* 950 to *m*/*z* 5000 and with a laser raster step size of 150 µm. Spectra from 500 laser shots were used to produce an average spectrum. The mass spectrometer was calibrated with a peptide mixture (Leu-enkephalin, Angiotensin II, Fibrinopeptide B, ACTH fragment (1–24), and insulin chain B). Ion images were generated using FlexImaging software (Version 4.1, Bruker Daltonics, Billerica, MA, USA) using RMS normalization.

To measure the accurate masses of the signals observed in IMS data in the cortex and nucleus regions, tissue from the cortex (within 400 µm from the capsule) and inner nucleus region (within 500 µm from the center) of a 55 Y.O. (C_5_1) cataract lens and the cortex region of 54 Y.O. (N_5_1) normal transparent lens was captured by laser capture microdissection (LCM). The LCM procedure was conducted using a PALM UV Laser MicroBeam laser capture microdissection system (Carl-Zeiss, Oberkochen, Germany) on tissue sections of 12 µm thickness on glass slides. The capture was done using a laser pressure catapulting (LPC) method and 37% laser energy. An area of roughly 1 mm^2^ was collected for each sample. The samples were collected in 20 µL of water and mixed with acetonitrile containing 2%TFA to final acetonitrile concentration of 5% and loaded on C18 SpinTips (Thermo Scientific, Rockford, IL, USA). The samples were washed with 0.1%TFA and eluted in 2.5 µL of 70% acetonitrile containing 0.1%TFA. The samples were mixed with 2.5 µL of saturated α-cyano-4-hydroxycinnamic acid (CHCA) matrix (Sigma-Aldrich, St Louis, MO, USA) in 70% acetonitrile (0.1%TFA) and masses were measured with high accuracy (<5 ppm) in a Bruker SolariX 15T FT-ICR mass spectrometer (Bruker Daltonics, Billerica, MA, USA) equipped with a Smartbeam II 2 kHz Nd:YAG (355 nm) laser. The mass spectrometer was externally calibrated prior to analysis using a peptide mixture (Leu-enkephalin, Angiotensin II, Fibrinopeptide B, ACTH fragment (1–24), and insulin chain B). The mass spectrometer was operated in positive ion mode and data were collected from *m*/*z* 700 to 4500 at a mass resolution of ~155,000 at *m*/*z* = 2360.2393.

### 2.3. Peptide Identification and LC-MS/MS Analysis

To identify the peptides that correspond to the low molecular weight signals in MALDI imaging results, peptides were extracted from the nucleus region of two cataract lenses using 20% acetonitrile, 80% water as described previously [17]. The samples were diluted using 0.1% formic acid to reduce the acetonitrile concentration to 3% and loaded onto a C18 trap column (50 mm × 150 μm) packed with Phenomenex Jupiter resin (5 µm mean particle size, 300 Å pore size). The trap column was connected with a fused silica capillary analytical column (150 mm × 100 µm) packed with Phenomenex Jupiter resin (3 µm mean particle size, 300 Å pore size). Peptides were analyzed by LC-MS/MS on a LTQ Orbitrap Velos (Thermo Scientific, San Jose, CA, USA) mass spectrometer coupled to an Eksigent NanoLC system. Mobile phase solvents consisted of 0.1% formic acid in water (solvent A) and 0.1% formic acid in acetonitrile (solvent B). Peptides were gradient eluted at a flow rate of 500 nL/min using the following gradient: 2–45%B in 40 min; 45–90%B in 15 min; 90%B for 2 min; 90–2%B in 1 min; and 2%B for 15 min (column equilibration). Peptides were introduced into the LTQ Orbitrap Velos using a nanoelectrospray source. The instrument was operated using a combined method consisting of both data-dependent electron transfer dissociation (ETD) and targeted ETD scan events. MS1 acquisition was performed in the Orbitrap (R = 60,000) with an AGC target value of 1e6. The MSn AGC target value of 9e5 was used in the Orbitrap and 2e4 in the ion trap. Dynamic exclusion (repeat count 1, exclusion list size 500, and exclusion duration 15 s) was enabled. For ETD, an isolation width of 3 *m*/*z* and reaction time of 80 ms were used. The ETD reagent ion AGC target was 3e6. ETD MS/MS spectra were interpreted manually.

## 3. Results

### 3.1. Detection of Low Molecular Signals in Cataract Nucleus

In this study, an untargeted approach using imaging mass spectrometry (IMS) was used to examine protein degradation in different regions of normal and cataract human lenses. In addition to intact crystallins and longer truncated crystallin fragments reported previously [9], IMS analysis also revealed lower molecular weight (LMW) signals and some of these signals were only detected in the nucleus region of nuclear cataract lenses. A typical IMS result is shown in Figure 1, which shows the presence of signals at *m*/*z* 2933 and 4098 in the nucleus of the two cataract lenses (55 Y.O. (C_5_1) and 56 Y.O. (C_5_2)). These two signals were not present in the nucleus of a 54 Y.O. (N_5_1) normal transparent lens.

Note that Figure 1 also showed the presence of a signal at *m*/*z* 2933 in the cortex of both normal and cataract lenses. Due to absence of this signal in the inner cortex and outer nucleus region, we hypothesized that the signal in the cortex could represent a different peptide with similar molecular weight to the one in the nucleus. Due to the low mass resolving power of the linear MALDI-TOF instrument used in this experiment, two molecules with close molecular weights, i.e., nominal isobars, could not be mass resolved. To test this hypothesis, tissues from the cortex and nucleus regions were collected by LCM and the samples were analyzed using a high mass resolution FT-ICR mass spectrometer. The results shown in Figure 2 confirmed our hypothesis. The protonated ion in the 54 Y.O.(N_5_1) cortex region has measured *m*/*z* value of 2930.4939 and the measured *m*/*z* value in 55 Y.O. (C_5_1) cataract lens cortex was 2930.4999, but the signal in the nucleus region has measured *m*/*z* value of 2931.6001. This result confirmed the 2933 signal in the cortex region shown Figure 1 corresponds to a different peptide from the signal in the cataract nucleus. The signal at *m*/*z* 2930.4939 in the lens cortex has not been identified in subsequent LC-MS/MS experiments of nuclear extracts; however, the mass matches a βB2-crystallin 182–205 peptide with a mass error 4.2 ppm and this βB2-crystallin peptide was previously reported to be present in human lenses [13].

### 3.2. Identification of Signals Detected by IMS and Further Confirmation of Cataract Specificity

Identification of the peptides at *m*/*z* 2933 and 4098 was performed based on their accurate masses and their tandem mass spectra in LC-MS/MS experiments. To identify these signals, a 20% acetonitrile extract from the nucleus region of a 70 Y.O. (C_7_1) cataract lens was analyzed by LC-MS/MS. The base peak chromatogram from the LC-MS/MS analysis is shown in Figure 3 which further confirms the presence of two abundant signals with the measured mass of a protonated ion ([MH]+) of 2931.5942 and 4096.1988. These measured masses match γS-crystallin peptide sequences 154–178 and 145–178, respectively, within 1 ppm mass accuracy. Their identifications were further confirmed by their tandem mass spectra. The ETD tandem mass spectrum of γS-crystallin peptide 154–178 (+4 ion) is shown in Figure 4A displaying excellent matches to the peptide sequence with a series of c and z ions. The tandem mass spectrum for γS-crystallin peptide 145–178 (+6 ion) can be found in Figure 4B. In addition to these two γS peptide signals, manual analysis of the major LC-MS/MS signals confirmed the presence of multiple γS-crystallin, βA3-crystallin and γD-crystallin C-terminal peptides in the cataract lens nucleus sample. These signals are labeled in the base peak chromatogram (Figure 3). A list of these peptides with their measured and predicted masses is shown in Table 1.

To further confirm whether γS-crystallin peptides 154–178 and 145–178 are indeed cataract-specific, several IMS datasets were acquired from multiple human lenses. The data are shown in Appendix A. The results from multiple datasets, acquired on different instruments and using different experimental conditions, consistently confirmed the presence of γS-crystallin 154–178 and γS-crystallin 145–178 in cataract lens nuclei, but not in normal lenses. A weak signal of γS-crystallin 145–178 can sometimes be detected in aged normal lens (70 Y.O. (N_7_1) and 78 Y.O. (N_7_2)), but the signal intensity of this peptide was significantly stronger in age-matched cataract lens nuclei. Moreover, a mild nuclear cataract could have been missed in the evaluation of these aged lenses.

IMS data were also acquired in a low mass range using a higher mass resolution reflector MALDI-TOF instrument. The results can be found in Figure 5. Consistent with aforementioned results, γS-crystallin 154–178 and γS-crystallin 145–178 were detected in cataract nuclei, but not in cataract cortex and normal lenses. Other than γS-crystallin 145–178 and γS-crystallin 154–178, other signals that were highly abundant in the cataract nucleus compared to cortex tissue in cataract and normal lenses were also detected (Figure 5). The signal at *m*/*z* 1204 was identified as γS-crystallin 169–178. Other signals at *m*/*z* 1849 and *m*/*z* 1779 correspond to αA-crystallin 66–80 and αA-crystallin 67–80, respectively. Consistent with previous reports [13,14,18], a weak signal of αA 66–80 and 67–80 was detected in the normal lens, but intensities of these peptides were significantly higher in the cataract lenses. The accurate mass measurement again showed the signal at nominal *m*/*z* 1389 corresponds to two peptides γS-crystallin 167–178 and γD-crystallin 162–173. As shown in Appendix A, the signal in the cortex was mainly from γS-crystallin 167–178 (calculated *m*/*z* 1388.7645), but in the nucleus, intensities of both γS-crystallin 167–178 and γD-crystallin 162–173 (calculated *m*/*z* 1388.8121) were similar. Therefore, both peptides contribute roughly equally to the signal at *m*/*z* 1389 in the nucleus. The weak signal of γS-crystallin 167–178 in normal lens and cataract lens cortex was also consistent with previous reports [13,14,19].

### 3.3. Low Molecular Weight Signals in Normal Lenses

As shown in Figure 3, some strong LMW signals are identified as βA3-crystallin peptides. Unlike γS-crystallin peptides, IMS analysis showed βA3-crystallin peptides were detected in both normal and cataract lenses. IMS results from three middle-aged lenses are shown in Figure 6. These results suggest that βA3-crystallin undergoes degradation and this process starts earlier than γS-crystallin degradation since strong signals from βA3-crystallin 188–215 and βA3-crystallin 189–215 can be detected in the cortical fiber cells of the middle-aged lenses. Signals from βA3-crystallin fragments did not show a difference between normal lens cortex and cataract lens cortex, however, signals from the longer peptides, especially βA3-crystallin 188–215, dramatically decreased and the signal from the shorter fragments (βA3-crystallin 191–215 and 192–215) significantly increased in the nucleus of the cataract lenses. The longest βA3-crystallin C-terminal peptide detected was 187–215 and shorter peptides were detected corresponding to removal of residues from the new N-terminus. These results suggest that initial cleavage occurs at the Asp186-His187 peptide bond, that further degradation of peptide products continues with age, and that this process is elevated in the cataract nucleus.

## 4. Discussion

Accumulation of low molecular weight (LMW) crystallin fragments in aged and cataract lenses has been characterized by multiple investigators [12,13,14,17,18] and some major crystallin fragments, including peptides from αA-, αB- and βA3-crystallins [13,14,15,18], have been repeatedly reported. However, with the exception of γS-crystallin 167–178 and 2–22 peptides, other γS-crystallin LMW peptides have not been reported. In this study, we identify LMW peptides from the γS-crystallin C-terminus as the dominating signals in the nucleus region of nuclear cataract lenses. Most noteworthy, these LMW fragments are cataract-specific since they were either not detected or were of very low abundance in normal lenses by IMS. One might ask why these abundant γS-crystallin peptides in cataract lens nucleus were not detected previously [13,14]. We speculate that the reason is due to the strong membrane binding and hydrophobic properties of these peptides. As previously suggested, γS-crystallin 167–178 can only be extracted from the membrane with NaOH or ethanol [18]. Thus, the peptides reported in this study are likely to be strongly bound to fiber cell membranes. In fact, the γS-crystallin 167–178 peptide has been reported to strongly bind to the cell membrane and may also affect water permeability [18]. Thus, the hydrophobic properties of C-terminal γS-crystallin peptides could induce protein aggregation and/or plasma membrane binding.

γS-crystallin, formerly known as βS-crystallin, belongs to the βγ-crystallin superfamily of lens crystallins. Based on sequence homology and predicted tertiary structure, γS-crystallin was found to be more closely related to the monomeric γ-crystallins than to the oligomeric β-crystallins and its name was changed to γS-crystallin [20]. γS-Crystallin is synthesized postnatally and is highly abundant in the ocular lens [21], whereas other γ-crystallins are primarily synthesized before birth and are mainly present in the lens nucleus [21]. Multiple single-residue mutations in γS-crystallin were reported to be associated with formation of different types of cataract [22]. Deamidation of Asn76 in γS-crystallin was found in greater abundance in cataract lenses compared to normal lenses [7]. The finding of the LMW γS-crystallin peptides to be specifically present in the nucleus of nuclear cataract lenses further supports the importance of γS-crystallin during cataractogenesis. Consistent with our results, it was reported that the monomeric full length of γS-crystallin was diminished in nuclear cataract lenses [23]. Clearly, extensive degradation of γS-crystallin occurs in nuclear cataract. How the properties of the observed C-terminal fragments contribute to opacification remains to be elucidated.

Our finding of cataract-specific LMW fragments in nuclear cataracts support the hypothesis that the development of cataract is due, at least in part, to the aggregation of crystallin fragments generated by the breakdown of crystallin proteins [15,24]. The discovery of a LMW αA-crystallin peptide, residues 66–80, that induced protein aggregation and light scattering as well as caused loss of α-crystallin chaperone activity [15] also supports this hypothesis [13,15]. We observed this specific αA-crystallin peptide in cataractous lens nuclei along with αA-crystallin 67–80. In addition to LMW peptides, truncated crystallins have been found to be more prone to aggregation [18].

Considering the potential protein aggregation properties of the LMW fragments, it is important to understand the mechanisms that mediate the cleavage of crystallins. No protease has been linked with previously reported LMW peptides [13,14]. Human βA3-crystallin was reported to possess autodegradative serine protease activity that could be responsible for degradation of the βA3-crystallin N-terminus (Gln4-Ala5 and Lys17-Met18) [25], however, this would not explain the formation of the βA3-crystallin C-terminal fragments. The longest βA3-crystallin C-terminal fragment is βA3-crystallin 187–215 corresponding to cleavage at Asp186. We believe the peptide 187–215 is formed first followed by further removal of residues from the N-terminus resulting in multiple peptides with different lengths. Our IMS results also confirm that the formation of βA3-crystallin peptides starts in young cortical fiber cells and the peptides formed were further degraded in the nucleus. Similarly, the formation of γD-crystallin 162–173 corresponds to cleavage at Asn 161. Spontaneous cleavage at Asp and Asn through a succinimide intermediate [26] or Glu or Gln through a glutarimide [27] is a well-known mechanism for protein cleavage and this process is also coupled with deamidation, isomerization, and racemization of Asn or Gln residues [26]. These mechanisms explain truncation, deamidation and isomerization at many different Asn, Asp, Gln and Glu residues throughout the lens proteome, especially the ones that are followed by small, flexible residues such as glycine [26,28,29]. Another spontaneous cleavage mechanism, i.e., cleavage on the N-terminal side of Ser that may involve an intein-like mechanism [19,30] could be responsible for the formation of γS-crystallin 167–178. Both proposed mechanisms are expected to occur in both normal and cataract lenses and, correspondingly, predicted LMW peptides can be detected in both normal and cataract lenses.

The two major cataract-specific γS-crystallin peptides detected in this study are produced by cleavage after Asn144 and Asp153. Therefore, similar to other lens peptides, spontaneous cleavage through a succinimide intermediate could also contribute to the formation of these peptides. Additional removal of N-terminus residues generates other shorter γS-crystallin peptides detected in nucleus of cataract lenses. However, the succinimide mechanism is a common mechanism that is responsible for the most prevalent modifications such as deamidation and isomerization in aged normal lenses [4,8,31]. It is questionable why these γS-crystallin peptides were only detected in cataract lens nuclei if a common aging mechanism is responsible for the formation of these peptides. In addition, succinimide mediated deamidation and truncation is characterized by formation of multiple isomers after deamidation and truncation. From our LC-MS/MS analysis of trypsin digested human lens samples, multiple isomers can be detected for the βA3-crystallin 178–193 (containing Asp186) and γD-crystallin 154–163 (containing Asn161); however, the γS-crystallin peptide 132–146 (plus deamidation) and 149–154 were present as single peaks indicating lack of isomerization (data not shown). Even though cleavage at Asn144 and Asp153 in γS-crystallin occurs at Asp and Asn residues, this cleavage may be through a different mechanism other than through a succinimide intermediate. Another possible explanation is that LMW peptides formed in normal lenses can be effectively degraded, but this degradation process is lost in cataract lenses resulting in accumulation of these peptides. Further study is needed to understand the mechanism for formation and accumulation of LMW γS-crystallin fragments in cataract lenses.

## Figures and Tables

**Figure 1 cells-11-04042-f001:**
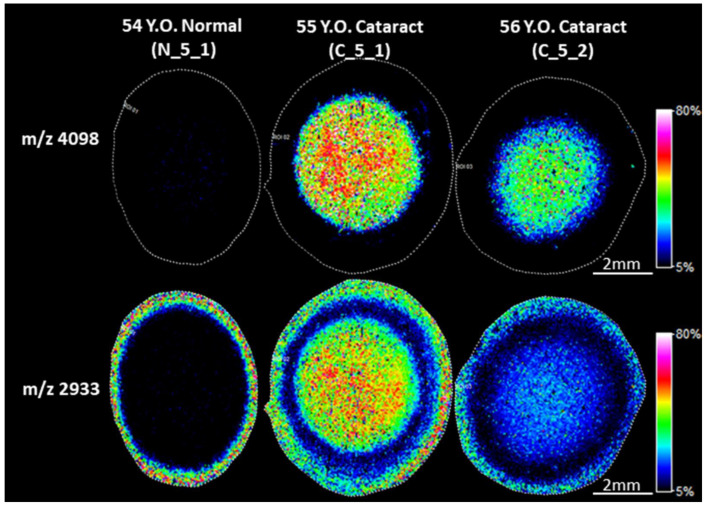
Cataract-specific signals in the nucleus of cataract lenses. Imaging mass spectrometry (IMS) analysis of human lens sections was performed on Bruker Rapiflex Tissuetyper using a 60 µm raster step size. IMS analysis identified two signals with *m*/*z* values of 4098 and 2933 that were only detected in the nucleus of cataract lenses, but not in normal lens nucleus. Accurate mass measurement confirmed *m*/*z* 2933 signal in the cortex region represent a different peptide than that represented by the signal in the cataract nucleus. Signal represents peak *m*/*z* value ± 0.1% m/z unit.

**Figure 2 cells-11-04042-f002:**
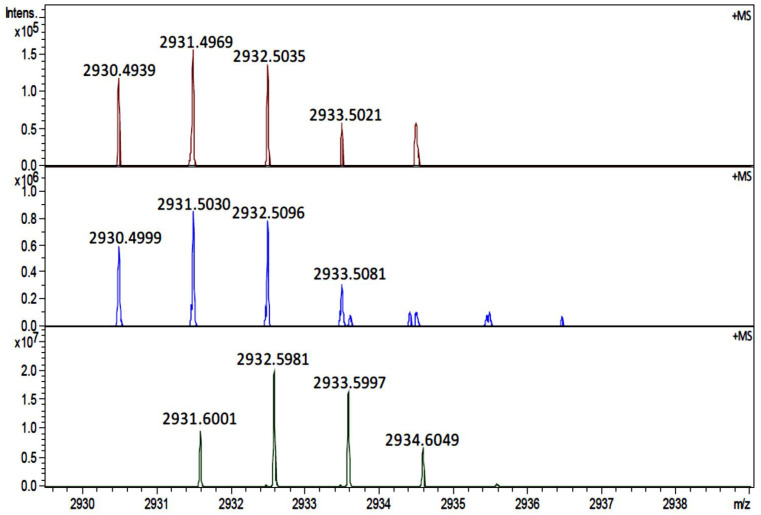
Accurate mass measurement of the signal at *m*/*z* 2933 in cortex and nucleus regions using an FT-ICR mass spectrometer. The positive mode mass spectra from LCM captured cortex and nucleus samples were zoomed in to show the isotopic distribution of signals at *m*/*z* 2933. The protonated ion in a 54 Y.O. (N_5_1) normal lens cortex region (**top row**) has a measured monoisotopic mass of 2930.4939 Da and 2930.4999 Da in 55 Y.O. (C_5_1) cataract lens cortex (**middle row**), but has a measured monoisotopic mass of 2931.6001 Da in 55 Y.O. cataract lens nucleus (**bottom row**).

**Figure 3 cells-11-04042-f003:**
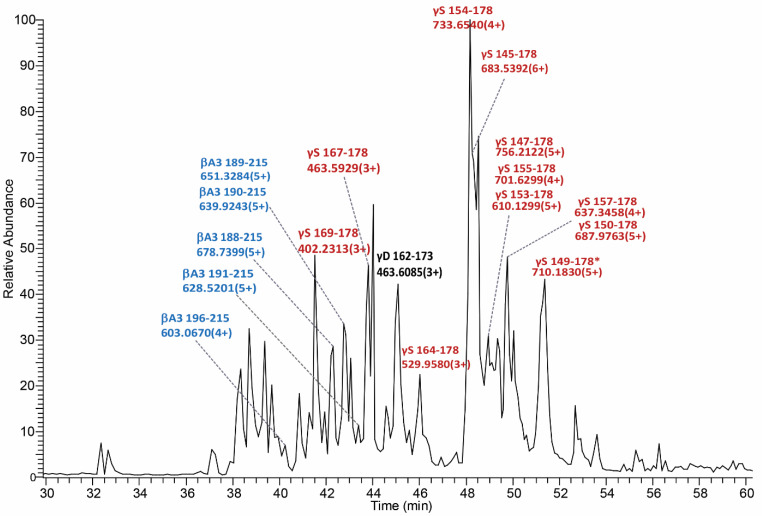
The base peak chromatogram of LC-MS/MS analysis of the extract from a 70 Y.O. (C_7_1) nucleus of a cataract lens. The base peak chromatogram is dominated by strong signals of γS- and βA3-crystallin fragments. Two strongest signals are identified as γS154–178 and γS145–178. Other base peak signals were identified as γS-crystallin C-terminal peptides (red), βA3-crystallin C-terminal peptides (blue) and a γD-crystallin C-terminal peptide (black). * indicates pyroGlu at the N-terminus.

**Figure 4 cells-11-04042-f004:**
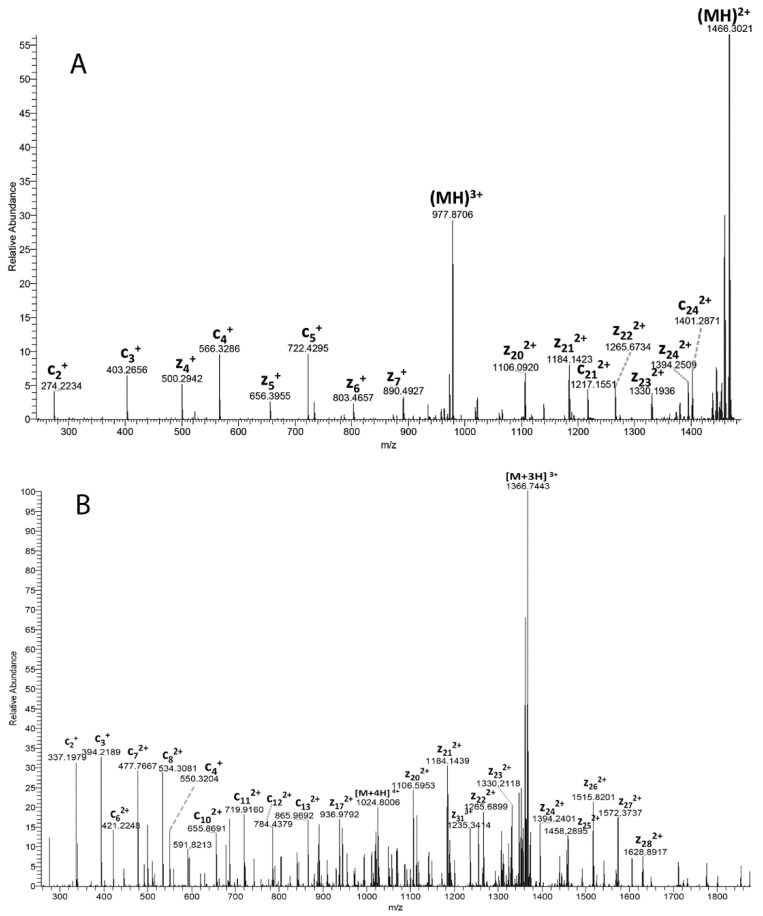
ETD tandem mass spectra of: (**A**) γS-crystallin 154–178 (+4 charged ion) and (**B**) γS-crystallin 145–178 (+6 charged ion). All masses labeled represent the monoisotopic masses.

**Figure 5 cells-11-04042-f005:**
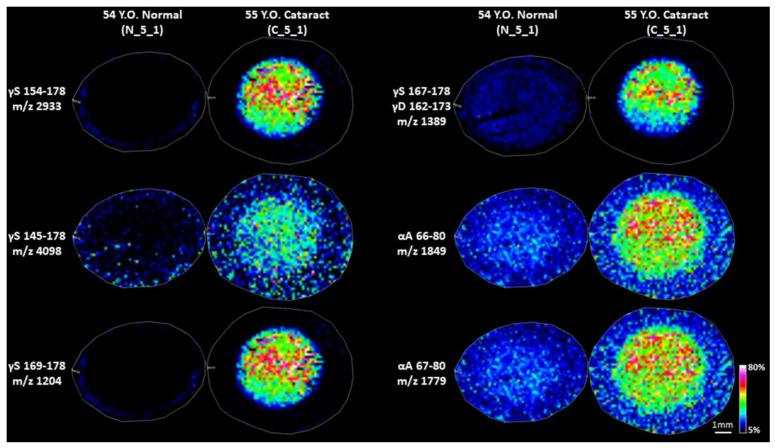
IMS images of LMW fragments specific or highly enriched in cataract lenses. Images were acquired on Bruker Rapiflex Tissuetyper using a 150 µm raster step size. Signal represents peak *m*/*z* value ± 0.01% *m*/*z* unit.

**Figure 6 cells-11-04042-f006:**
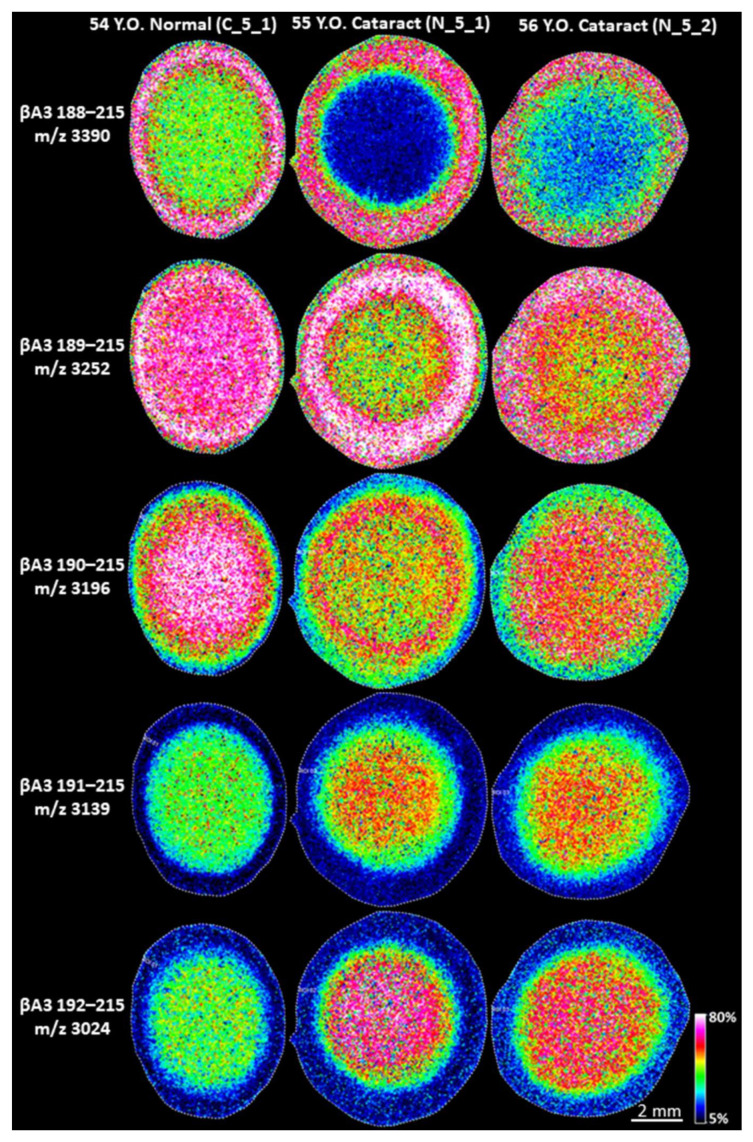
IMS analysis of βA3 peptides in normal and cataract lenses. IMS analysis of human lens sections was performed on Bruker Rapiflex Tissuetyper using a 60 µm raster step size. Multiple βA3-crystallin fragments were detected in normal and cataract lenses. The protein cleavage started in lens cortex and βA3 fragments were further degraded and their formation was accelerated in cataract lens nuclei. Signal represents peak *m*/*z* value ± 0.1% *m*/*z* unit.

**Table 1 cells-11-04042-t001:** Identified major γS-, γD- and βA3-crystallin peptides in the nucleus of a 70 Y.O. (C_7_1) cataract lens.

Protein	Predicted *m*/*z* ^a^	Observed *m*/*z*	Error (ppm)
Gamma S 145–178	4096.1992	4096.1988	0.1
Gamma S 147–178	3777.0347	3777.0319	0.74
Gamma S 149–178 (pyroGlu)	3546.8856	3546.8859	0.1
Gamma S 150–178	3435.8536	3435.8524	0.35
Gamma S 153-178	3046.6221	3046.6204	0.56
Gamma S 154–178	2931.5952	2931.5942	0.35
Gamma S 155–178	2803.5002	2803.4978	0.87
Gamma S 157–178	2546.3627	2546.3614	0.52
Gamma S 164–178	1587.8602	1587.8594	0.47
Gamma S 167–178	1388.7645	1388.7641	0.25
Gamma S 169–178	1204.6797	1204.6793	0.29
Beta A3 188–215	3389.6747	3389.6704	1.3
Beta A3 189–215	3252.6158	3252.6129	0.89
Beta A3 190–215	3195.5943	3195.5924	0.6
Beta A3 191–215	3138.5729	3138.5714	0.5
Beta A3 196–215	2409.2494	2409.2462	1.3
Gamma D 162–173	1388.8121	1388.8109	0.83

^a^ monoisotopic *m*/*z*.

## Data Availability

The raw data is available on request that should be send to k.schey@vanderbilt.edu.

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
