# Peer review of "Imaging Cataract-Specific Peptides in Human Lenses"

_cells, 2022, doi:10.3390/cells11244042_

Round 1

Reviewer 1 Report

The authors of manuscript “Imaging Cataract-Specific Peptides in Human Lenses” Kevin L. Schey, Zhen Wang, Kristie L Rose and David M.G. Anderson studied the spatial localization of protein fragments in the human lens with imaging mass spectrometry. Some crystallin fragments were found to be colocalized with nuclear clouding of the lens. The authors supposed the hydrophobic properties of C-terminal γS-crystallin peptides could induce protein aggregation and therefore cataract formation. At the same time, the authors point out directly that the putative pathways of partial proteolysis of crystallins are nonspecific and can be present in both transparent and cloudy lenses. In addition, we could not forget that “post hoc ergo propter hoc” another words, the increased level of γS-crystallin peptides in central lens region could be result but not cause of cataract formation. So results obtained need a further investigation.

 Taking in account that investigation demonstrates novel insights in cataractogenesis which eligible for “Cells” journal the manuscript “Imaging Cataract-Specific Peptides in Human Lenses” could be published after corrections needed.

Listed below are my specific notes on the manuscript.

Line 13 - 14 “Consistent with previous reports, the pro-aggregatory a-crystallin 66-80 peptide as well as αA-crystallin 67-80…”

Remark: Santhoshkumar et al., studied the aA-crystallin only, a-crystallin was not investigated. 

Line 53 “Matherials and Methods”

Remark: the lenses used should be described in more detail. Especially, the method used to measure the degree of lens opacification should be stated. Photos of lenses, both transparent and cloudy, are desirable to provide.

Line 132 “…cataract lens (stage 5-6) than in 56yo 132 cataract lens (stage 3-4)…”

Remark: clarification is needed. Please see above remark.

Line 165

Remark: what was the cataract grade of 78yo cataract lens? Clarification is needed.

Reviewer 2 Report

This paper describes, using various mass spectrometry methods, the presence of previously undescribed 154-178 and 145-178 gamma-S crystallin peptide fragments in the nucleus of cataract lenses.  The authors also describe progressive degradation of βA3 crystallin peptides from ~188-215 to ~190-215 lengths. 

The main concern I have is how many lenses were actually studied? It seems that the bulk of the data relate to just three lenses - a 54 Y.O. normal, and two cataract lenses (at 55 and 56 Y.O). In supplemental figure 1, five other normal lenses (7, 23, 37, 70 and 78 Y.O.) and two 70 Y.O. cataract lenses are mentioned.  However, it is hard to know what overlap there is between the two 70 Y.O cataract samples in this figure and similarly in Supplemental Fig. 2, two cataract lenses at 55 Y.O. each are listed as opposed to the 55 and 56 Y.O samples in Fig. 1.   A table listing all lenses examined with details of age, sex, cataract types and severity etc is needed and ideally this should provide unique identifiers so that readers can discern whether the same or different lenses are being presented in the various analyses.  Ideally some quantitative/statistical assessments to determine how consistent/significant these nuclear peptides are among nuclear cataracts and if there really is a correlation with stage of cataract.

While generally well written there are various aspects that could be improved for the non-specialist reader. In particular, the methods sections are impenetrable to the non-specialist. Please provide enough information that non-specialists (like this reviewer) can understand the context, rationale for settings, standards and the purpose of the various protocols. While this can be gleaned eventually from reading the Results, it is less evident from reading the methods.

Minor comments

P2, line 46: Define HBO

P2, line 55: Define abbreviation NDRI (National Disease Research Interchange as opposed  to National Drug Research Institute in Australia!).

P2, line 56: Institution affiliation for Dr Garland?

P2, line 58: Lower case ‘c’ for Cryostat? Why is LEICA capitalized?

P2, line 63:’was then sprayed’ - matrix is singular. Use 'were' if matrices were sprayed

P2, line 66 (and elsewhere): Is this the correct symbol for degrees and not a superscript 'o'?

P2, line 77: Inconsistent spacing before units such as um throughout the manuscript.  It should be noted that units such as mass, volume and length are preceded by a space, whereas units such as % and degrees are not. Please provide units for mass range (daltons) in line 77.

P2, line 82: Please define m/z - presumably mass to charge ratio?  Why this specific value? Need to specify that the IMS reveals signals at various m/z values?  It currently appears that this specific ratio was selected prior to analysis and not identified as result of analysis.

P2, line 84: Inconsistent spacing before yo - see above. I believe the correct abbreviation for years -old is Y.O.?

P3, line 132: Authors describe a correlation between intensity of signal and stage of cataract. Was this borne out in other cataracts at different stages? Suggest including the cataract stage in the figure or at least in a table with all other samples.  How well characterized were these cataract samples?

Figure 1: The way Figure 1 is presented it seems the authors already knew these were gamma-S crystallin fragments.  However, this only becomes apparent until later in the Results after the LCMS/MS analyses.  Please clarify this in the text or the legend and define gamma-S in the figure. Alternatively, perhaps this might be addressed by combining Figs 1 and 2 and removing the gammaS peptide labels from Figure 1?

P4, line 143: Authors hypothesize that cortical signal was a different peptide – just some explanation for this. Presumably because of lack of signal in the deeper cortex?

P5, line 168: Please define [MH]+ - proton ionised form?

P8, line 194: How many lenses?  Tendency to not detail the number of lenses examined.

Supp. Fig.1: Can the authors explain the slightly different m/z values for the crystallin fragments in the IMS images with different machines?  Similarly, why is there a weak signal in the 7 Y.O. lens with the Autoflex apparatus?

P8, line 200: This comment about lack of cataract identification is somewhat concerning as to how well any of these samples were characterized clinically.

Round 2

Reviewer 2 Report

The authors have responded appropriately to all comments raised by this reviewer.